

# Differences in urinary incontinence symptoms and pelvic floor structure changes during pregnancy between nulliparous and multiparous women

Dan Luo[1,*], Ling Chen[1,*], Xiajuan Yu[1,2], Li Ma[3], Wan Chen[3], Ning Zhou[3] and Wenzhi Cai[1]

[1] Department of Nursing, Shenzhen hospital of Southern Medical University, Shenzhen, Guangdong, China
[2] Department of Neonatology, Shenzhen Maternity & Child Health Care Hospital, Shenzhen, Guangdong, China
[3] Department of Ultrasound, Shenzhen Hospital of Southern Medical University, Shenzhen, Guangdong, China
[*] These authors contributed equally to this work.

Corresponding author
Wenzhi Cai, kfzywyh@126.com

## ABSTRACT

**Background**. This study was performed to compare changes in urinary incontinence (UI) symptoms and pelvic floor structure during pregnancy between nulliparous and multiparous women.

**Methods**. A cross-sectional survey was performed among pregnant women from July 2016 to January 2017. In total, 358 pregnant women from two hospitals underwent an interview and pelvic floor transperineal ultrasound assessment. A questionnaire regarding sociodemographic, gynecological, obstetric features and the International Consultation on Incontinence Questionnaire-Short Form (ICIQ-SF) were used for the interview. Imaging data sets were analyzed offline to assess the bladder neck vertical position (BNVP), urethral angles ($\alpha$, $\beta$, and $\gamma$ angles), and hiatal area (HA) at rest and at maximal Valsalva maneuver (VM).

**Results**. After excluding 16 women with invalid data, 342 women were included. The prevalence ($\chi^2 = 9.15$, $P = 0.002$), frequency ($t = 2.52$, $P = 0.014$), usual amount of UI ($t = 2.23$, $P = 0.029$) and scores of interference with daily life ($t = 2.03$, $P = 0.045$) during pregnancy were higher in multiparous than nulliparous women. A larger bladder neck descent (BND) ($F = 4.398$, $P < 0.001$), HA ($F = 6.977$, $P < 0.001$), $\alpha$ angle ($F = 2.178$, $P = 0.030$), $\beta$ angle ($F = 4.404$, $P < 0.001$), and $\gamma$ angle ($F = 2.54$, $P = 0.011$) at VM were discovered in pregnant women with UI than without UI. Multiparous women had a significantly higher BND ($t = 2.269$, $P = 0.024$) and a larger $\alpha$ angle ($F = 2.894$, $P = 0.004$), $\beta$ angle ($F = 2.473$, $P = 0.014$), and $\gamma$ angle ($F = 3.255$, $P = 0.001$) at VM than did nulliparous women.

**Conclusion**. Multiparous women experienced more obvious UI symptoms and pelvic floor structure changes during pregnancy than did nulliparous women.

## INTRODUCTION

In 2002, the International Continence Society (ICS) defined urinary incontinence (UI) as "the involuntary passage of urine for any reason" (*Abrams et al., 2002*). UI is a common condition during pregnancy, with a prevalence of 16.8%–72.0% (*Adaji et al., 2011*; *Bo et al., 2012*; *Tanawattanacharoen & Thongtawee, 2013*; *Rincon, 2015*; *Abdullah et al., 2016*). UI may exert a negative effect on pregnant women's working routines, free time activities, and sexual intimacy (*Wijma et al., 2001*). *Dolan et al. (2003)* found that the risk of UI doubled 15 years after the development of UI during a woman's first pregnancy.

However, studies of UI during pregnancy have mainly focused on nulliparous women; a limited number of epidemiological studies have targeted multiparous women (*Hvidman et al., 2002*; *Raza-Khan et al., 2006*; *Scarpa et al., 2006*; *Wesnes et al., 2007*; *Al-Mehaisen et al., 2009*). In a systematic review of the epidemiology of UI during pregnancy, the estimated incidence of UI in pregnancy ranged from 45% to 54% among multiparous women but only 28% to 45% among nulliparous women (*Wesnes, Hunskaar & Rortveit, 2012*). Since implementation of the "Two-Child Policy" in China, the number of multiparous women is expected to increase. Thus, UI during pregnancy in multiparous women should be taken seriously.

The pathophysiology of UI during pregnancy involves pregnancy-associated pelvic floor changes (*Clement et al., 2013*). Previous studies using transperineal ultrasound to assess changes in the pelvic floor anatomy during pregnancy in nulliparous women revealed that the predominant changes involved the levator hiatal dimensions and the position and mobility of the bladder neck (BN) (*Dietz, 2004*; *Shek, Kruger & Dietz, 2012*; *Van Veelen, Schweitzer & Van der Vaart, 2014a*; *Chan et al., 2014*). Although it is believed that multiparous women have a higher prevalence of UI than nulliparous women, the differences in the severity of UI and the changes in the pelvic floor structure during pregnancy between nulliparous and multiparous women remain unclear. Therefore, the aim of this study was to assess and compare the differences in the symptomatology of UI during pregnancy and the morphology of the pelvic floor anatomy between nulliparous and multiparous women to provide a scientific basis for further research in prenatal care.

## MATERIALS AND METHODS

### Ethics statement

Ethics approval for the study was granted by the Ethics Committee of Shenzhen Materity & Child Health Care Hospital (2016–30). All participants provided written informed consent.

### Participants

Three hundred fifty-eight women with a singleton pregnancy who were undergoing routine antenatal assessment in the obstetric clinics of various hospitals of Shenzhen from July 2016 to January 2017 were recruited. The exclusion criteria were (1) a history of UI before pregnancy; (2) a history of abdominal and vaginal surgery; (3) diabetes and hypertension; and (4) placenta previa, threatened abortion, amniotic fluid abnormalities, fetal growth restriction, or vaginal bleeding. The elimination criteria were (1) a failure to complete all investigation content and (2) unreliable pelvic ultrasound data.

An interview of the patients was conducted by XJ Y to investigate the patients' age, gestational weeks, body mass index (BMI), constipation during pregnancy, number of pregnancies, prior abortions/miscarriages, and delivery history. UI was diagnosed using the International Consultation of Incontinence Questionnaire-Short Form (ICIQ-SF). This questionnaire can also be used to evaluate the severity of UI. It comprises three scored items (Questions 1–3), frequency of UI (score range, 0–5), usual amount of UI (score range, 0–6), interference with everyday life (score range, 0–10), and a self-diagnostic item (Question 4, not scored) (*Avery et al., 2004*). This instrument is recommended by the ICS and has been validated in China (*Huang et al., 2008*).

## Transperineal ultrasound assessment

After completing the interview covering the patients' general information and symptoms of UI, transperineal ultrasound examination was performed using a Voluson E8 system (GE Healthcare, Chicago, IL, USA) with a 5–9 MHz three-dimensional autosweep transducer. The pregnant women should empty urine before examination and then lay in the supine position. The probe was smeared with a coupling agent and then covered with a condom. A vaginal probe was placed between the two labia, directly below the urethral orifice. The symphysis pubis was set as the axis point indication, and the median sagittal plane of the vagina, urethra, and anal canal was then obtained. Volume scanning was performed, the selection box was properly adjusted, and the BN, urethra, vagina, proximal anal canal, distal anal canal, and pubic symphysis were scanned. Two- and three-dimensional cross-sectional sonograms of the pelvic floor in the resting position and at maximal Valsalva maneuver (VM) were collected. The VM process lasted for approximately 5 s until satisfactory images were obtained. Ultrasound assessment was performed by three experienced investigators (LM, NZ, and WC) who were blinded to all other information. During the examination, the probe remained closely attached to the perineum to prevent air interference and ensure that the quality of the picture was not affected.

Offline analysis of the imaging data sets was conducted using the GE Kretz 4D View version 10.3 software program (GE Healthcare). All data sets were analyzed by one of two investigators (DL or LC) who were blinded to the UI symptoms. The reference line was a horizontal line placed at the inferoposterior margin of the symphysis pubis. The following parameters were measured at rest and VM: (1) bladder neck vertical position (BNVP): distance (in cm) between the BN and the inferior-posterior margin of the symphysis pubis. A positive value indicated that the BN was positioned cranial of the symphysis pubis and a negative value that is was positioned caudal of the symphysis pubis. Bladder neck descent (BND) is the difference of BN between VM and rest (*Dietz, 2011*); (2) α angle: the angle between the proximal urethra and the trigone (*Volløyhaug et al., 2017*); (3) β angle: the angle between the bladder neck-symphysis line and the middle of the symphysis (*Pregazzi et al., 2002*); (4) γ angle: the angle between the proximal urethra and the horizontal line (*Volløyhaug et al., 2017*); (5) hiatal area (HA) was measured as the area (in cm$^2$) bordered by the pubovisceral muscle, pubic symphysis and the inferior pubic ramus (*Van Veelen, Schweitzer & Van der Vaart, 2014a*), (Fig. 1).

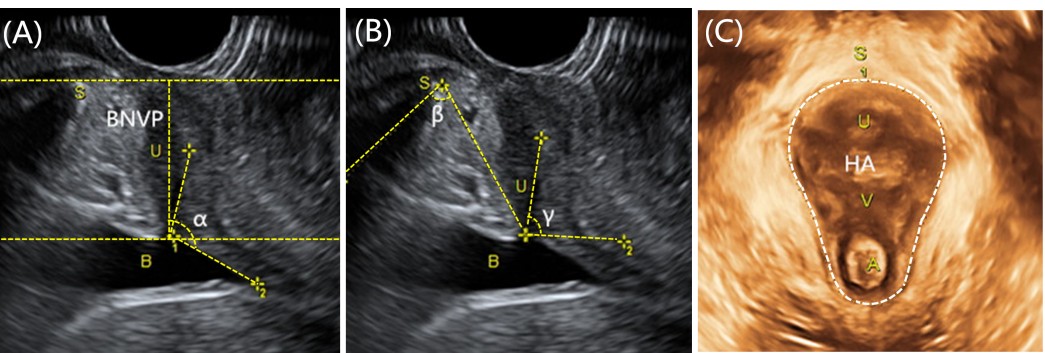

**Figure 1  Transperineal ultrasound measurement of pregnant women.** (A) the measurment of BNVP and α angle; (B) the measurment of β angle and γ angle; (C) the measurment of HA. B, bladder; S, symphysis pubis; U, urethra; V, vagina; A, anus; BNVP, bladder neck vertical position; HA, hiatal area.

## Statistical analysis

SPSS 13.0 (SPSS Inc., Chicago, IL, USA) and GraphPad Prism 5.0 (GraphPad; San Diego, CA, USA) were used for all statistical analyses. Descriptive statistics were presented as mean ± standard deviation for continuous variables and as frequency (percentage) for categorical variables. Student's $t$ test was used to analyze quantitative data. Chi-square test was used for categorical data. Covariance analyses (gestational weeks as the covariate) were conducted to compare the ultrasound parameters between pregnant women with and without UI and between nulliparous and multiparous women. All statistical tests were two-tailed. The level of significance was set at 0.05.

## RESULTS

### Comparisons of demographic characteristics between nulliparous and multiparous women

Ten pregnant women failed to perform the VM effectively (generally, HA should be larger at VM than at rest; the data were considered invalid if the HA results were exactly the opposite), and six did not finish the interview. Therefore, 342 pregnant women were included in our final data analysis. Among those who completed the study, 179 were nulliparous women and 163 were multiparous women. Table 1 shows the demographic characteristics of the study population and the associated factors between the nulliparous and multiparous groups. There were significant differences in age, number of pregnancies, and prior miscarriages/abortions between the nulliparous and multiparous women (Table 1).

### Comparison of UI severity during pregnancy between nulliparous and multiparous women

The results of the comparison of the prevalence and severity of UI between the nulliparous and multiparous women are shown in Fig. 2. Fifty-seven (31.8%) nulliparous women and 78 (47.9%) multiparous women were diagnosed with UI during pregnancy. The prevalence of UI was significantly different between these two groups ($\chi^2 = 9.15$, $P = 0.002$). The scores for the frequency of UI ((1.55 ± 0.56) vs. (1.21 ± 0.63)), usual amount of

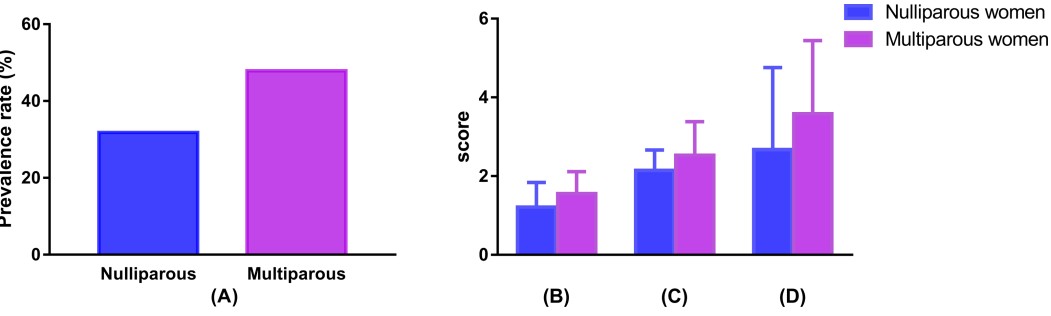

**Figure 2  Comparison of prevalence and severity of UI between nulliparous and multiparous women.**
(A) Comparison of prevalence rate of UI; (B) comparison of frequency of UI; (C) comparison of usual amount of UI; (D) comparison of interference with daily life. UI, urinary incontinence.

**Table 1  Comparison of demographic characteristics between nulliparous and multiparous women ($n = 342$).**

| | Nulliparous ($n = 179$) | Multiparous ($n = 163$) | $t/\chi^2$ | $P$ |
|---|---|---|---|---|
| Age (years) (mean ± SD) | 28.42 ± 3.82 | 32.71 ± 3.70 | 10.532[a] | <0.001[*] |
| Gestational weeks (mean ± SD) | 19.58 ± 9.57 | 19.30 ± 9.77 | 0.267 | 0.790 |
| Gestation $n$ (%) | | | | |
|    First trimester | 64 (35.8) | 58 (35.6) | | |
|    Second trimester | 60 (33.5) | 57 (35.0) | 0.099[b] | 0.952 |
|    Third trimester | 55 (30.7) | 48 (29.4) | | |
| BMI (mean ± SD) | 23.17 ± 3.52 | 23.18 ± 3.37 | 0.026[a] | 0.979 |
| Constipation during pregnancy $n$ (%) | | | | |
|    Yes | 49 (27.4) | 51 (31.3) | 0.632[b] | 0.476 |
| Number of pregnancies (mean ± SD) | 1.35 ± 0.53 | 2.79 ± 0.85 | 19.020[a] | <0.001[*] |
| Prior miscarriages/abortions $n$ (%) | | | | |
|    Yes | 57 (31.8) | 87 (53.4) | 16.224[b] | <0.001[*] |

Notes.
[a] $t$.
[b] $\chi^2$.
[*] Significant difference between nulliparous and multiparous women.
BMI, body mass index.

UI(($2.53 \pm 0.85$) vs. ($2.14 \pm 0.52$)), and interference with daily life (($3.58 \pm 1.86$) vs. ($2.67 \pm 2.09$)) were significantly higher in multiparous than nulliparous women (all $P < 0.05$).

## Comparison of pelvic floor ultrasound parameters between pregnant women with and without UI

The results of the covariance analysis of pelvic floor ultrasound parameters (gestational weeks as the covariate) are shown in Table 2. Pregnant women with UI showed a larger β angle and HA than those without UI at rest. The α angle, β angle, γ angle, BNVP and HA at VM were significantly different between pregnant women with and without UI. Larger differences in β angle, BNVP, and HA between VM and rest were found in pregnant women with UI than without UI. The α angle and γ angle at rest and between VM and rest as well as the BNVP at rest were not significantly different between the two groups (Table 2).

**Table 2   Comparison of pelvic floor ultrasound parameters between pregnant women with and without UI ($n = 342$).**

|  | UI ($n = 135$) | Non-UI ($n = 207$) | F | P |
|---|---|---|---|---|
| **R** |  |  |  |  |
| α angle (°) | 120.11 ± 15.63 | 118.53 ± 16.19 | 1.711 | 0.192 |
| β angle (°) | 66.94 ± 13.26 | 62.19 ± 11.43 | 11.271 | 0.001[*] |
| γ angle (°) | 80.09 ± 8.59 | 78.10 ± 8.53 | 2.928 | 0.088 |
| BNVP (cm) | 2.59 ± 0.51 | 2.66 ± 0.48 | 0.833 | 0.362 |
| HA (cm$^2$) | 11.64 ± 2.51 | 10.65 ± 1.98 | 12.250 | 0.001[*] |
| **VM** |  |  |  |  |
| α angle (°) | 141.02 ± 16.44 | 137.07 ± 16.36 | 3.970 | 0.047[*] |
| β angle (°) | 105.84 ± 28.88 | 93.91 ± 21.16 | 15.429 | <0.001[*] |
| γ angle (°) | 113.95 ± 15.60 | 110.03 ± 12.74 | 4.406 | 0.037[*] |
| BNVP (cm) | 1.01 ± 1.03 | 1.47 ± 0.73 | 17.005 | <0.001[*] |
| HA (cm$^2$) | 15.82 ± 3.53 | 13.24 ± 3.23 | 37.930 | <0.001[*] |
| **VM-R** |  |  |  |  |
| α angle (°) | 20.92 ± 12.18 | 18.54 ± 14.25 | 0.815 | 0.367 |
| β angle (°) | 38.90 ± 27.01 | 31.72 ± 18.94 | 5.977 | 0.015[*] |
| γ angle (°) | 33.94 ± 14.72 | 31.93 ± 13.58 | 1.187 | 0.277 |
| BNVP (cm) (BND) | 1.59 ± 0.97 | 1.19 ± 0.69 | 14.459 | <0.001[*] |
| HA (cm$^2$) | 4.18 ± 2.85 | 2.59 ± 2.31 | 24.279 | <0.001[*] |

**Notes.**
[*]Significant difference between pregnant women with and without UI.

UI, urinary incontinence; R, at rest; VM, maximal Valsalva maneuver; VM-R, the difference of VM and rest; BNVP, bladder neck vertical position; BND, bladder neck descent; HA, hiatal area.

## Comparison of pelvic floor ultrasound parameters during pregnancy between nulliparous and multiparous women

Table 3 shows the covariance analysis results of pelvic floor ultrasound parameters (gestational weeks as the covariate) between nulliparous and multiparous women. The α angle, β angle, γ angle, and BNVP during pregnancy were significantly different at VM between nulliparous and multiparous women. Multiparous women showed a larger α angle and γ angle between VM and rest than did nulliparous women. The BND was significantly different between nulliparous and multiparous women. However, the HA was not significantly different between nulliparous and multiparous women.

## DISCUSSION

Multiparous women experienced more severe symptoms of UI than did nulliparous women. The rate of UI in nulliparous women was 31.8%, which was significantly lower than that in multiparous women (47.9%). Our results are basically consistent with those of previous studies (*Wesnes et al., 2007*; *Lin et al., 2014*). UI, described as "social cancer", has been listed as one of the five most common chronic diseases worldwide according to the World Health Organization and remains a global problem affecting women of all ages and across different cultures and races (*Minassian, Drutz & Al-Badr, 2003*). In the present study, we discovered a higher frequency of UI, the larger usual amount of UI, and

**Table 3** Comparison of pelvic floor ultrasound parameters between nulliparous and multiparous women ($n = 342$).

| | Nulliparous ($n = 179$) | Multiparous ($n = 163$) | $F$ | $P$ |
|---|---|---|---|---|
| **R** | | | | |
| $\alpha$ angle (°) | 118.81 ± 14.98 | 119.54 ± 17.03 | 0.155 | 0.694 |
| $\beta$ angle (°) | 62.99 ± 13.37 | 65.25 ± 11.13 | 2.904 | 0.089 |
| $\gamma$ angle (°) | 78.44 ± 9.62 | 79.38 ± 7.31 | 1.112 | 0.292 |
| BNVP (cm) | 2.63 ± 0.50 | 2.65 ± 0.48 | 0.170 | 0.680 |
| HA (cm$^2$) | 10.99 ± 2.25 | 11.10 ± 2.26 | 0.251 | 0.616 |
| **VM** | | | | |
| $\alpha$ angle (°) | 136.20 ± 15.83 | 141.31 ± 16.82 | 8.494 | 0.004[*] |
| $\beta$ angle (°) | 95.43 ± 23.49 | 102.12 ± 26.48 | 6.506 | 0.011[*] |
| $\gamma$ angle (°) | 109.25 ± 12.62 | 114.14 ± 15.10 | 11.074 | 0.001[*] |
| BNVP (cm) | 1.37 ± 0.81 | 1.19 ± 0.96 | 3.970 | 0.047[*] |
| HA (cm$^2$) | 14.13 ± 3.63 | 14.40 ± 3.51 | 0.668 | 0.414 |
| **VM-R** | | | | |
| $\alpha$ angle (°) | 17.39 ± 12.28 | 21.77 ± 14.43 | 10.008 | 0.002[*] |
| $\beta$ angle (°) | 32.44 ± 20.42 | 36.87 ± 24.85 | 3.478 | 0.063 |
| $\gamma$ angle (°) | 30.82 ± 12.32 | 34.82 ± 15.51 | 7.144 | 0.008[*] |
| BNVP (BND) (cm) | 1.25 ± 0.74 | 1.46 ± 0.91 | 5.635 | 0.018[*] |
| HA (cm$^2$) | 3.14 ± 2.64 | 3.30 ± 2.67 | 0.420 | 0.517 |

**Notes.**

[*]Significant difference between nulliparous and multiparous women.

UI, urinary incontinence; R, at rest; VM, maximal Valsalva maneuver; VM-R, the difference of VM and rest; BNVP, bladder neck vertical position; BND, bladder neck descent; HA, hiatal area.

more serious interference with daily life in multiparous than nulliparous women. With the implementation of the "Two-Child Policy" in China, the growing number of multiparous women has drawn attention to the screening and prevention of UI during pregnancy.

Most previous studies of the relationship between UI during pregnancy and pelvic structure merely took nulliparous women into consideration. In addition to the higher prevalence of UI and more severe UI symptoms among multiparous than nulliparous women, our study also revealed that multiparous women experienced more obvious changes in their pelvic structure than did nulliparous women. Urethral support plays a critical role in the pathogenesis of UI. The BN position, BN mobility, and HA can be regarded as objective indicators of the supportive capacity of the pelvic floor, which can be measured effectively by transperineal ultrasound (*Dietz, 2004*; *Majida et al., 2009*; *Shek & Dietz, 2010*; *Siafarikas et al., 2013*; *Jundt et al., 2010*). The $\alpha$ angle, $\beta$ angle, $\gamma$ angle, and BNVP indicate the BN position. Differences in these parameters between VM and rest represent BN mobility; larger differences indicate more obvious BN activity and weaker pelvic floor support. Levator ani muscle trauma may lead to a larger width of the HA, particularly its anterior part (*Svabik, Shek & Dietz, 2009*), which has been found to be associated with the development of UI (*Chan et al., 2017*). A larger $\beta$ angle and HA both at rest and VM; a larger $\alpha$ angle, $\gamma$ angle, and BNVP at VM; and a larger BND have also been found in pregnant women with UI than without UI, even after adjustment for the

influence of gestational weeks, which is important because UI symptoms are more likely to occur in the last than first trimester of pregnancy (*Wesnes, Hunskaar & Rortveit, 2012*). All of these pelvic floor parameters (α angle, β angle, γ angle, HA, and BNVP) have shown good intraclass correlation coefficients in previous studies (*Pregazzi et al., 2002*; *Chan et al., 2014*; *Naranjo-Ortiz et al., 2016*). Our results are similar to those of previous studies. *Chan et al. (2014)* found that a greater BND increased a woman's likelihood of developing UI during pregnancy. *Al-Saadi (2016)* found that women with UI had a larger α angle and β angle at VM than did women without UI. *Sendag et al. (2003)* reported that women with UI had a larger α angle and γ angle at VM than did women without UI. A study by *Van Veelen, Schweitzer & Van der Vaart (2014b)* showed significant differences in the HA at rest and at VM between pregnant women with and without UI. We found that the BND, HA, β angle, and γ angle at VM were greater in women with UI than without UI, confirming the results of previous studies.

Multiparous women experienced more obvious pelvic changes than did nulliparous women in the present study. Multiparous women had larger α, β, and γ angles and a larger BNVP at VM as well as a larger BND than did nulliparous women during pregnancy after adjustment for the influence of gestational weeks. As mentioned above, the α angle, β angle, γ angle, and BNVP at VM and the BND have been found to differ between pregnant women with and without UI; thus, we were not surprised to find that multiparous women experienced more significant pelvic floor changes than nulliparous women during pregnancy.

Multiparous women also showed higher age, larger numbers of pregnancies, and a higher prevalence of prior miscarriages/abortions than did nulliparous women in our study, which might explain why multiparous women were more susceptible to UI and pelvic floor changes than nulliparous women. The previous studies reported that higher age leads to the loss of nerve function and a decrease in the total number of striated muscle fibers of the urethral sphincter at a rate of 2% per year and a gradual decline in the maximum urethral closure pressure of approximately 15% per decade (*Pandit et al., 2000*; *Wesnes & Lose, 2013*). A previous study demonstrated that pregnancy itself has an effect on the pelvic floor (*Dietz et al., 2004*); the mechanical and hormonal effects of pregnancy can lead to biomechanical, neurological, or neuromuscular changes to the pelvic floor and pelvic organ support (*South et al., 2009*; *Chen et al., 2005*), which may lead to more obvious BN mobility and contribute to pelvic floor dysfunction. *Brown et al. (2010)* investigated 1,507 pregnant women and found that prior miscarriages/abortions was a significantly high risk factor for UI during pregnancy; our results are consistent with these findings. However, our data require a further cohort study to verify the reasons why multiparous women developed more severe UI symptoms and pelvic floor structure changes during pregnancy than did nulliparous women.

## LIMITATIONS

One of the limitations of our study is its cross-sectional design. To elucidate the pathophysiology of delivery-induced UI, women should ideally be examined before,

during, and after their pregnancy to assess preexisting differences in the pelvic floor anatomy and the absence of UI before, during, and after pregnancy (*Chan et al., 2014*). Additionally, we could not eliminate the possibility that some patients were unable to push downward to the extent required for the VM, which may have resulted in underestimation of the actual pelvic floor parameters.

## CONCLUSIONS

Our results indicate that multiparous women had a higher prevalence of UI and more severe UI symptoms than did nulliparous women, which could be associated with weaker pelvic floor support.

### Funding

This work was supported by the Shenzhen Baoán Science and Technology Innovation Project (No. 2016CX313) and the Southern Medical University Nursing Project (No. Z2016005). The funders had no role in study design, data collection and analysis, decision to publish, or preparation of the manuscript.

### Grant Disclosures

The following grant information was disclosed by the authors:
Shenzhen Baoán Science and Technology Innovation Project: 2016CX313.
Southern Medical University Nursing Project: Z2016005.

### Competing Interests

The authors declare there are no competing interests.

### Author Contributions

- Dan Luo and Ling Chen conceived and designed the experiments, performed the experiments, analyzed the data, wrote the paper, prepared figures and/or tables.
- Xiajuan Yu conceived and designed the experiments, performed the experiments, contributed reagents/materials/analysis tools, wrote the paper.
- Li Ma, Wan Chen and Ning Zhou performed the experiments, contributed reagents/materials/analysis tools, reviewed drafts of the paper.
- Wenzhi Cai conceived and designed the experiments, reviewed drafts of the paper.

### Human Ethics

The following information was supplied relating to ethical approvals (i.e., approving body and any reference numbers):

The Shenzhen Maternity & Child Health Care Hospital granted ethical approval to carry out the study within its facilities (Ethical Application Ref: 2016-30).

### Data Availability

The raw data has been uploaded as Data S1.

## Supplemental Information

Supplemental information for this article can be found online at http://dx.doi.org/10.7717/peerj.3615#supplemental-information.

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
