# Peer review of "Differences in urinary incontinence symptoms and pelvic floor structure changes during pregnancy between nulliparous and multiparous women"

_PeerJ, doi:10.7717/peerj.3615_

## Round 0.1 · original submission · Major Revisions

Dear Authors,

The Reviewers found your manuscript very interesting and recommended publication after a revision including both minor and major comments.

I would suggest to take into consideration the Reviewers' comments and discuss or apply them to your manuscript in order to reach the level requested for publication.

Best regards

·

Basic reporting

• Basic Reporting
- The English language should be overall improved to ensure that your international audience can clearly understand your text. I suggest that you have a native English speaking colleague review your manuscript.
- Raw data properly supplied. However data sheet is not complete (lack of some demographic characteristics)
- Abstract – Method: please correct or remove number of patients (340). I see you recruited 358 pts and 342 was considered for analysis. Please specify each statitstical result next to considered parameters instead of collecting all togethere at the end of each sentence.
- Intro and background well referenced
- Method
- VM abbreviation should be also speficied in the full text and noy only in the abstract
 Figure 1 Please improve legend. You may also better enlight referral lines
 Please specify Bladder Neck Descent evaluation in the methods
- Line 109-111 this consideration in improper in M&M, i think it could be added in the discussion. Please also add which parameters are you considering.
- Result
- Typo line 136: “index”
- Figure 2 Please improve legend (2.1 is not explained). Also consider relative prevalence instead of absolute prevalence to improve graph 2.1
- Please consider condensing your result in table/tables
- Please mark with * significant results in Figures 3-4
- Discussion
 Please add discussion and references about bladder neck mobility differences between parous and nulliparous women
 Please specify what your study can add to the current literature
- References: please double check, some references are listed two times

Experimental design

Experimental design
- Interview and questionnaire were collected at heterogeneus gestational weeks. This is an important issue, since UI sympoms are more likely to occur in the last weeks of pregnancy compared to the first ones.
- Same for the ultrasound evalutation. This may represent a confounding factor since ultrasound parameters change during pregnancy. See for istance the following paper already in your reference list.
van Veelen GA, Schweitzer KJ, van der Vaart CH.Ultrasound imaging of the pelvic floor: changes in anatomy during and after first pregnancy.Ultrasound Obstet Gynecol. 2014 Oct;44(4):476-80. doi: 10.1002/uog.13301.

Validity of the findings

• 3. Validity of the findings
- I see distribution of gestation trimester is similar but this may be a little broad considering the previous issues with evaluation time point. Is it possible to specify mean+/-SD gestation week? Specific data are not reported on supplied database.

Additional comments

Thank you for the oppurtunity to review this interesting paper. This study is aimed to compare urinary incontinence symptoms and pelvic floor ultrosound aspects between nulliparous and multiparous women during pregnancy. This paper confirms current literature findings about differences in UI prevalence and ultrasound aspects between nulliparous and multiparous women. However there are some issue related to the paper that should be improved/clarify
1-english language should be improved
2-clarify some methodological issues
d3-iscussion must enlight strenghts and innovations of the study.

Reviewer 2 ·

Basic reporting

The article is well written using a clear English with unambiguous words.
The introduction and background are adequate and characterized by literature appropriately referenced, which offers to the reader a complete background about the urinary incontinence during pregnancy and morphology of the pelvic floor anatomy.
The article is well structured in an acceptable format with an adequate use of high-resolution figures and explanatory graphs, which render the text more intelligible.
Raw data are available, clear and well categorized.

Experimental design

The aim of this study is coherent with the aims and scope of the Journal. The authors present a corss-sectional study with aim to assess and compare the differences of symptomatology of the urinary incontinence during pregnancy and morphology of pelvic floor anatomy between nulliparous and multiparous women, and provide scientific basis for further research in the prenatal care. Although the issue treated is a well-known pregnancy problem and the cross-sectional study is limiting, the authors offer a point of view that can contribute to improve the knowledge about the urinary incontinence pathophysiology during pregnancy.
The investigation was rigorously conducted and the ethical standards were respected. The Ethics approval for the study and its number was reported. Moreover, the authors declare that a written informed consent was provided to all participants.
Methods and the transperineal ultrasound technique by which the study was developed are clearly described and so reproducible.

Validity of the findings

The urinary incontinence during pregnancy is not an original topic but a point of view that derives from the Chinese population before of “two-child Policy” could be useful in order to better evaluate changings in the future.
The results are robust and the statistical analysis is coherent to the aim of the study. Strengths and limitations are clearly reported and discussed.
Nevertheless, in order to make the paper more readable, I suggest that you limit to report in the text the results already described in the graphs. I would limit to describe in the text only the more significant results.

---

## Round 0.2 · accepted · Accept

Dear Authors,

I would like to compliment with you for the efforts provided in addressing the Reviewers' comments.

All Reviewers fell that your manuscript has reached the level of publication and can be accepted in its current form.

Best regards

Salvatore Andrea Mastrolia

·

Basic reporting

Basic reporting
- The English language was well improved
- Raw data properly supplied.
- Abstract clear
- Intro and background well referenced
- Method clear and well referenced
- Result clear
- Discussion improved with strenghts and innovations, well referenced
- References: correct

Experimental design

Experimental design: clear

Validity of the findings

Validity of the findings
Gestational weeks issue clarified

Additional comments

Comments for the author
Thank you for the oppurtunity to review this interesting paper. This study is aimed to compare urinary incontinence symptoms and pelvic floor ultrosound aspects between nulliparous and multiparous women during pregnancy. I think this paper can be a precious adding to understand differences in UI prevalence and ultrasound aspects between nulliparous and multiparous women.

Reviewer 2 ·

Basic reporting

Ok.

Experimental design

Ok.

Validity of the findings

Ok.

Additional comments

I suggest to accept the paper as it is.